# Exploiting Dynamic Vector-Level Operations and a 2D-Enhanced Logistic Modular Map for Efficient Chaotic Image Encryption

**DOI:** 10.3390/e25081147

**Published:** 2023-07-31

**Authors:** Hongmin Li, Shuqi Yu, Wei Feng, Yao Chen, Jing Zhang, Zhentao Qin, Zhengguo Zhu, Marcin Wozniak

**Affiliations:** 1Key Laboratory of Hunan Province on Information Photonics and Freespace Optical Communications, Hunan Institute of Science and Technology, Yueyang 414006, China; lihminnew@sina.com (H.L.);; 2College of Physics and Electronics, Hunan Institute of Science and Technology, Yueyang 414006, China; 3School of Mathematics and Computer Science, Panzhihua University, Panzhihua 617000, China; chenyaoscu@outlook.com (Y.C.); zjpzh@tom.com (J.Z.); qinzhentao@pzhu.edu.cn (Z.Q.);; 4Faculty of Applied Mathematics, Silesian University of Technology, 44-100 Gliwice, Poland

**Keywords:** chaotic system, image encryption, hyperchaotic map, dynamic vector-level operations, chaotic performance evaluation, security analysis

## Abstract

Over the past few years, chaotic image encryption has gained extensive attention. Nevertheless, the current studies on chaotic image encryption still possess certain constraints. To break these constraints, we initially created a two-dimensional enhanced logistic modular map (2D-ELMM) and subsequently devised a chaotic image encryption scheme based on vector-level operations and 2D-ELMM (CIES-DVEM). In contrast to some recent schemes, CIES-DVEM features remarkable advantages in several aspects. Firstly, 2D-ELMM is not only simpler in structure, but its chaotic performance is also significantly better than that of some newly reported chaotic maps. Secondly, the key stream generation process of CIES-DVEM is more practical, and there is no need to replace the secret key or recreate the chaotic sequence when handling different images. Thirdly, the encryption process of CIES-DVEM is dynamic and closely related to plaintext images, enabling it to withstand various attacks more effectively. Finally, CIES-DVEM incorporates lots of vector-level operations, resulting in a highly efficient encryption process. Numerous experiments and analyses indicate that CIES-DVEM not only boasts highly significant advantages in terms of encryption efficiency, but it also surpasses many recent encryption schemes in practicality and security.

## 1. Introduction

In the current era of massive multimedia information, enormous volumes of multimedia data are constantly being produced and then circulated through various channels, such as the Internet and the Internet of Things (IoT). Among various forms of multimedia data, digital images are the most frequently employed because they possess the capability to intuitively and efficiently communicate information [1]. Significantly, in the present open network environment, it is urgent to protect these rapidly spreading images more efficiently and securely. Otherwise, catastrophic consequences, such as privacy leakages, may occur [2]. As commonly acknowledged, data encryption is a relatively direct and effective way of safeguarding data. However, because images possess some inherent features different from text, such as strongly correlated pixels, traditional encryption algorithms are not well-suited for encrypting image data [3]. Consequently, image encryption research utilizing new technologies and methods has been increasingly attracting attention from researchers due to various reasons, such as privacy protection, commercial security, and military security [4]. In the past few years, to offer more efficient and secure protection for image data, lots of new image encryption algorithms or schemes have been continuously suggested [5,6,7,8,9,10]. For these newly proposed image encryption algorithms or schemes, this paper will collectively refer to them as encryption schemes hereafter.

Due to their intrinsic characteristics, such as parameter sensitivity and ergodicity, which coincidentally meet the construction requirements of cryptosystems, chaotic systems are utilized in almost one-third of non-traditional encryption schemes for image security [11,12]. Here, we can list some examples of recently proposed representative schemes. In [13], Pourasad et al. developed a chaos-based encryption scheme with wavelet transforms. By employing chaotic sequences, they diffused the input image first and then performed wavelet transformation and confusion operations on the diffused image. Finally, they obtained the ciphertext image through inverse wavelet transformations. In [14], Xian et al. constructed a logistic map-based encryption scheme exploiting fractal sorting matrices. Their scheme directly adopted the hash output of the input image to create key components and then employed two consecutive rounds of scrambling and one round of XOR diffusion to complete image encryption. In [15], a chaos-based scheme based on image splitting was suggested by Kamal et al. This scheme first divided the input image into blocks and then introduced zigzag scan, rotation and block permutation to achieve the pixel scrambling. After being XORed with a chaotic matrix, the final ciphertext image was generated to prevent possible unauthorized access. By employing a two-dimensional (2D) logistic-sine map (2D-LSM), Hua et al. [16] designed an image encryption scheme based on Latin squares. Their scheme incorporates point-to-point pixel scrambling and cross-plane diffusion to complete image encryption. In [17], Li et al. suggested an image encryption scheme based on DNA operations, which exploits non-adjacent blocks and permutation blocks to scramble the input image and then employs dynamic bidirectional diffusion to obtain the final ciphertext image. Similarly, considering the substitution effect of dynamic DNA encoding, two image encryption schemes using scrambling and diffusion architectures were proposed successively [18,19]. In [20], Feng et al. also developed an image encryption scheme based on image filtering and discrete logarithm, in which image filtering can diffuse a large number of pixels at the same time, and discrete logarithm transformation exerts the encryption effect of pixel substitution. With the aid of the superb randomness provided by chaotic sequences, the previously mentioned schemes, along with other recently proposed ones, have exhibited rather effective encryption outcomes and have successfully passed various common security tests [21,22].

Given the limitations of classical chaotic systems, there are also many researchers dedicated to creating novel chaotic systems that can better fulfill the requirements of image encryption [16,21,23,24,25,26,27,28,29]. In [23], Hua et al. suggested a two-dimensional (2D) modular chaotification system (2D-MCS) to enhance the chaotic performance of existing maps. By introducing two coupling parameters and the modulo one transformation, Ablay [24] proposed a novel LE-enhanced chaotification model. This model can convert any two one-dimensional (1D) chaotic maps into 2D chaotic maps with uniform trajectory distributions and better chaotic performance. Similarly, by introducing a so-called buffeting parameter, Zhang et al. [25] suggested a buffeting chaotification model (BCM). In [26], based on the classic Hénon map, a 2D parametric polynomial chaotic system (2D-PPCS) was constructed. The simulation experiments show that the chaotic performance of 2D-PPCS is better than that of the Hénon map. By coupling the logistic map and cubic map, Nan et al. [28] developed a logistic coupling cubic chaotic map (2D-LCCCM). Significantly, although 2D-LCCCM achieves better chaotic performance than two seed maps, its structure is very complex, which is not conducive to engineering applications including image encryption.

As revealed by the latest cryptanalysis research results on chaotic image encryption, some encryption schemes still possess the following problems [30,31,32,33,34]. First, the chaotic performance of the exploited system is poor. For example, the chaotic range of the system is discontinuous, and the trajectory distribution is not uniform. Second, the composition of the secret key is unreasonable and poses practical problems. For instance, exploiting a hash value directly as a key component brings key management difficulties. Third, the design of the encryption process is not rigorous, resulting in security flaws or low encryption efficiency. Accordingly, to address the aforementioned shortcomings, we first constructed a two-dimensional enhanced logistic modular map (2D-ELMM) and then developed a chaotic image encryption scheme based on vector-level operations and 2D-ELMM (CIES-DVEM). In brief, our study brings the following contributions and novelties:(1)A robust 2D hyperchaotic map called 2D-ELMM is constructed, and its superiority is confirmed through reliable chaos performance metrics such as sample entropy (SE) and Kolmogorov entropy (KE).(2)Based on the newly constructed 2D-ELMM, a novel image encryption scheme called CIES-DVEM is developed, which incorporates dynamic vector-level operations that help improve encryption efficiency and enhance security.(3)Numerous simulation experiments and corresponding analyses demonstrate that our newly developed CIES-DVEM not only boasts remarkably high security but also exhibits a considerable advantage in terms of efficiency.

Our study is structured as follows for the remaining sections: In Section 2, 2D-ELMM is introduced in detail, and its performance is evaluated and compared by exploiting several chaos performance metrics. In Section 3, both the overall structure of CIES-DVEM and its individual encryption steps are elaborately described. In Section 4, numerous simulation experiments and corresponding analyses are presented to verify and highlight the security and efficiency superiorities of CIES-DVEM; and Section 5 concludes our study.

## 2. Proposed 2D-ELMM

This section gives the definition of 2D-ELMM and utilizes common chaos metrics such as Lyapunov exponent (LE), SE, and KE to evaluate its chaotic performance. Meanwhile, in order to demonstrate the superiorities of 2D-ELMM, its experimental results are also compared with five recently reported 2D chaotic maps.

### 2.1. Construction of 2D-ELMM

Classic chaotic maps like the logistic map and tent map have weaknesses such as uneven trajectory distributions and discontinuous chaotic intervals, which cannot sufficiently meet the needs of many applications, including image encryption. Moreover, compared with 1D chaotic maps, high-dimensional hyperchaotic maps usually have more complex chaotic behaviors. It should be pointed out that if a chaotic system has too many dimensions, such as three or four, it may cause efficiency problems that are unacceptable for many applications [4,35,36]. In recent times, an elevated number of researchers have developed 2D chaotic maps by exploiting a variety of methods, such as coupling, cascading, and combining, as illustrated in Table 1 [16,21,23,26,28,29].

However, there is another issue with these newly reported maps: namely, their structures are still too complex for certain engineering implementations. Considering this, we built a novel hyperchaotic map called 2D-ELMM on the basis of existing maps, which is presented below.
(1)$$\left\{\begin{array}{c} x_{k}=e^{a}x_{k-1}\left(e^{b}y_{k-1}-1\right)\;\mathrm{mod}\;1,\\ y_{k}=e^{b}y_{k-1}\left(e^{a}x_{k-1}-1\right)\;\mathrm{mod}\;1, \end{array}\right.$$
where $x_{k}$ and $y_{k}$ represent the two outputs produced during the *k*-th iteration of 2D-ELMM, while $x_{k-1}$ and $y_{k-1}$ indicate the corresponding two inputs. *a* and *b* are two control parameters adopted in 2D-ELMM. We designed these two parameters in an exponential form, so that the trajectory of 2D-ELMM can rapidly diverge, resulting in a more excellent chaotic performance. The purpose of modular operation is to confine the trajectory within a predetermined scope. A straightforward structure similar to the logistic map can be observed in 2D-ELMM, which is advantageous for engineering implementation and efficiency enhancement. In fact, the 2D-ELMM not only has a simpler structure but also exhibits outstanding chaotic performance, surpassing several newly reported leading chaotic maps.

### 2.2. Lyapunov Exponent

The indicator LE is extensively employed. By ascertaining if a system has an LE above 0, one can infer whether the system is in a state of chaos [37]. Figure 1 provides two 3D representations of LEs for 2D-ELMM. When a>1 and b>1, both LE values are greater than 0, which means that 2D-ELMM is hyperchaotic. In addition, these two values climb rapidly as *a* and *b* increase, and when (a,b)=(10,10), the values reach as high as 19.8770 and 10.0109, respectively. Consequently, when it comes to LE, 2D-ELMM exhibits outstanding hyperchaotic dynamics, which is exactly what numerous engineering applications, such as image encryption, demand.

Additionally, comparative experiments were also conducted. The suggested values from relevant literature were employed to configure the parameters in other maps during these experiments [16,21,28,29]. In 2D-ELMM, *b* is set to 10. The experimental outputs in Figure 2 show that 2D-ELMM has a continuous chaotic range, and both its LE values are noticeably higher than those of other maps. In other words, 2D-ELMM is superior to these most recent maps in regard to LE.

### 2.3. Bifurcation and Trajectory Diagrams

As a prominent graphical tool commonly employed to depict chaotic systems, bifurcation diagrams can visually show whether the system is in a chaotic state. When drawing bifurcation diagrams, researchers usually set different control parameters to track the state evolution of chaotic systems within a certain range of parameters [38]. Figure 3 presents the six bifurcation diagrams that we drew to confirm the trajectory distribution of 2D-ELMM. As we can see, regardless of how *a* and *b* are changed, 2D-ELMM produces results that are evenly spread across the entire value range. From the viewpoint of bifurcation diagrams, it can be concluded that 2D-ELMM exhibits remarkable chaotic behavior.

Like the bifurcation diagram, the trajectory diagram is also often employed by scholars to visually analyze the dynamics of a chaotic system. The output of a chaotic system is hoped to be evenly distributed throughout the overall output space for various purposes, such as image encryption. The six trajectory diagrams of 2D-ELMM can be found in Figure 4. Obviously, we obtained similar experimental results: that is, the trajectory of 2D-ELMM is extremely evenly distributed throughout the output space.

### 2.4. Sample Entropy

The self-similarity of time-series can be determined by sample entropy (SE). If the SE value of the chaotic sequence generated by a chaotic system is relatively high, then this system exhibits greater chaotic complexity [39]. Mathematically, we can calculate the SE value of a chaotic sequence *Q* by exploiting the following equation.
(2)SE(N,λ,d)=−ln(U(λ)/V(λ)),
where *N* is the length of *Q*, *d* represents the dimension, λ denotes the threshold for similarity comparison, and *U* and *V* are the numbers of the vectors that satisfy certain conditions. Through Equation (Equation 2), we assessed the SE performance of 2D-ELMM together with five other maps. Figure 5 illustrates the relevant experimental results. One can observe that 2D-ELMM’s SE values are significantly higher than those of other maps throughout the entire parameter range. This suggests that 2D-ELMM’s SE performance is considerably better in comparison to the other maps.

### 2.5. Kolmogorov Entropy

Kolmogorov entropy (KE) is an entropy capable of depicting the progression of a dynamical system. A dynamical system is chaotic, and its trajectory becomes unpredictable when it has a non-zero KE value. Furthermore, greater unpredictability and improved chaotic complexity are indicated by a relatively larger KE value. For a phase space with *d* dimensions, it is possible to divide it into a sequence of boxes $\left(a_{0},a_{1},\ldots ,a_{d}\right)$ measuring ε in size and establish the definition of KE in the following way:(3)$$KE=-\lim_{t\to 0}\lim_{\varepsilon \to 0}\lim_{d\to 0}d^{-1}t^{-1}\sum_{a_{0},a_{1},\ldots ,a_{d}}P(a_{0},a_{1},\ldots ,a_{d})\ln P\left(a_{0},a_{1},\ldots ,a_{d}\right).$$
In Equation (Equation 3), *t* represents the delay, and $P(a_{0},a_{1},\ldots ,a_{d})$ denotes the probability. In order to verify the KE performance of 2D-ELMM, we carried out experiments on 2D-ELMM and five other maps by leveraging the methodology presented in [40]. The relevant experimental results are depicted in Figure 6. It is easy to observe that similar to SE, the KE values of 2D-ELMM are considerably greater than those of the other five maps throughout the entire parameter range. Moreover, 2D-ELMM exhibits the highest level of stability in terms of KE.

## 3. CIES-DVEM

To provide more confirmation and demonstration of 2D-ELMM’s exceptional performance, a new image encryption scheme called CIES-DVEM was developed by exploiting 2D-ELMM. Figure 7 shows the encryption flowchart for CIES-DVEM. As depicted in Figure 7, CIES-DVEM consists of eight encryption steps, which are referred to as the generation of key streams, hash value stacking, dynamic binary diffusion (row diffusion), dynamic binary scrambling (column scrambling), dynamic binary scrambling (row scrambling), and dynamic binary diffusion (column diffusion). In subsequent subsections, we will describe these encryption steps in detail.

### 3.1. Generation of Key Streams

In our CIES-DVEM, a raw sequence $Q^{\left(0\right)}$ is first created by applying the secret key $K=\{x_{0},y_{0},a,b\}$. To put it differently, the initial step is to utilize *K* for iterating 2D-ELMM, thereby producing $Q^{\left(0\right)}$ of length
(4)L=M+N+2×α.
In Equation (Equation 4), *M* and *N* represent the number of rows and columns of the image that is to be encrypted, and α=M×N. Note that $Q^{\left(0\right)}$ is formed by interleaving the state values obtained by each iteration of 2D-ELMM; that is, $Q^{\left(0\right)}=\left(x_{1},y_{1},x_{2},y_{2},x_{3},y_{3},\ldots \right)$.

Then, $Q^{\left(0\right)}$ is converted into the key streams $Q^{\left(1\right)}$, $Q^{\left(2\right)}$, $Q^{\left(3\right)}$, $Q^{\left(4\right)}$, $Q^{\left(5\right)}$, and $Q^{\left(6\right)}$, which are required for each encryption step. Specifically, the conversion process for CIES-DVEM’s key streams is as follows.
(5)$$Q^{\left(1\right)}=\left\lfloor Q^{\left(0\right)}\left(1:M\right)\times 10^{15}\right\rfloor \;\mathrm{mod}\;256,$$
(6)$$Q^{\left(2\right)}=\left\lfloor Q^{\left(0\right)}\left(M+1:M+N\right)\times 10^{15}\right\rfloor \;\mathrm{mod}\;256,$$
(7)$$Q^{\left(3\right)}=\left\lfloor Q^{\left(0\right)}\left(M+N+1:M+N+\alpha \right)\times 10^{15}\right\rfloor \;\mathrm{mod}\;256,$$
(8)$$Q^{\left(4\right)}=\left\lfloor Q^{\left(0\right)}\left(M+N+\alpha +1:M+N+2\times \alpha \right)\times 10^{15}\right\rfloor \;\mathrm{mod}\;256,$$
(9)$$Q^{\left(5\right)}=Q^{\left(0\right)}\left(M+N+1:3\times M+3\times N\right),$$
(10)$$Q^{\left(6\right)}=Q^{\left(0\right)}\left(M+N+\alpha +1:3\times M+3\times N+\alpha \right),$$
where $\left\lfloor x\right\rfloor$ returns the largest integer less than *x*.

### 3.2. Hash Value Stacking

Previous studies have shown that the inability to resist plaintext attacks, especially chosen-plaintext attacks, is the most significant reason why many image encryption schemes are vulnerable to being cracked [30,31,32,33]. Accordingly, in order to increase CIES-DVEM’s plaintext sensitivity and enhance its ability to resist plaintext attacks, we perform an operation called hash value stacking on the plaintext image prior to executing scrambling and diffusion operations. Algorithm 1 gives the pseudocode for hash value stacking. In Algorithm 1, utilizing the 32 bytes of hash value $V^{\left(0\right)}$, we first determine the modular sum $v^{\left(s\right)}$ and byte-by-byte bitwise XOR result $v^{\left(x\right)}$ of these bytes. Next, two matrices $U^{\left(s\right)}$ and $U^{\left(x\right)}$ related to $V^{\left(0\right)}$ are constructed based on the key streams $Q^{\left(1\right)}$ and $Q^{\left(2\right)}$. Finally, the output $C^{\left(1\right)}$ of this encryption step is obtained by stacking $U^{\left(s\right)}$ and $U^{\left(x\right)}$ onto *P* in a modular addition manner.
**Algorithm 1** Hash value stacking algorithm**Input:** The plaintext image *P* and its SHA-256 hash value $V^{\left(0\right)}$ of length 32 bytes, and the key streams $Q^{\left(1\right)}$ and $Q^{\left(2\right)}$.  1:Set both $v^{\left(s\right)}$ and $v^{\left(x\right)}$ to 0;  2:**for**  i=1 to 32 **do**  3:   $v^{\left(s\right)}=\left(v^{\left(s\right)}+V^{\left(0\right)}\left(i\right)\right)\;\mathrm{mod}\;256$;  4:   $v^{\left(x\right)}=v^{\left(x\right)}⊕V^{\left(0\right)}\left(i\right)$;  5:**end for**  6:Set $U^{\left(x\right)}=transpose\left(Q^{\left(1\right)}\right)\times Q^{\left(2\right)}$;  7:Set $U^{\left(s\right)}=v^{\left(s\right)}\times U^{\left(x\right)}$;  8:$U^{\left(x\right)}=v^{\left(x\right)}\times U^{\left(x\right)}$;  9:Set $C^{\left(1\right)}=\left(P+U^{\left(s\right)}+U^{\left(x\right)}\right)\;\mathrm{mod}\;256$**Output:** The intermediate ciphertext image $C^{\left(1\right)}$.

### 3.3. Dynamic Binary Diffusion

Diffusion is a fundamental design principle that designers must adhere to when developing robust cryptosystems [41,42]. Notably, many new image encryption schemes have been broken due to weak diffusion processes. Actually, the fixed and single diffusion methods employed by these schemes are a significant factor contributing to this situation. In light of this, an encryption step called dynamic binary diffusion is devised in CIES-DVEM to overcome these shortcomings. A straightforward instance of dynamic binary diffusion is illustrated in Figure 8.

From this example, it can be seen that the input image is first logically and dynamically divided into two partitions. Then, these two partitions are diffused in two different ways. The left partition is diffused at the row level by employing modular addition, while the right partition is diffused in the form of bitwise XOR. Algorithm 2 gives the pseudocode for dynamic binary diffusion. In Algorithm 2, $\dot{+}$ represents modular addition with a modulus of 256, while ⊗ stands for the Hadamard product. The boundary of partitioning the input image $C^{\left(1\right)}$ is dynamically determined by $r^{\left(p\right)}$, which in turn depends on $v^{\left(x\right)}$ and $Q^{\left(3\right)}$. Since $v^{\left(x\right)}$ is a plaintext-related parameter obtained in hash value stacking, the dynamic binary partition helps to enhance plaintext relevance. Additionally, the two partitions are diffused in different ways, which further enhances the security of CIES-DVEM. Note that what we describe here is the row diffusion version of dynamic binary diffusion. Since the column diffusion version is highly similar, it is omitted here for brevity.
**Algorithm 2** Dynamic binary diffusion algorithm**Input:** The input image $C^{\left(1\right)}$, $v^{\left(x\right)}$ and $U^{\left(x\right)}$ obtained in hash value stacking, and the key streams $Q^{\left(3\right)}$.  1:Get the height *M* and width *N* of $C^{\left(1\right)}$;  2:Reshape $Q^{\left(3\right)}$ into a matrix of size M×N;  3:Set $r^{\left(p\right)}=\left(\left(v^{\left(x\right)}+Q^{\left(3\right)}\left(1,1\right)\right)\;\mathrm{mod}\;\left\lfloor N/8\right\rfloor \right)+\left\lfloor N/2\right\rfloor$;  4:Set $R^{\left(t\right)}=1:r^{\left(p\right)}$;  5:Set $C^{\left(2\right)}\in N^{M\times N}$;  6:$C^{\left(2\right)}\left(1,R^{\left(t\right)}\right)\;=\;C^{\left(1\right)}\left(1,R^{\left(t\right)}\right)\dot{+}Q^{\left(3\right)}\left(1,R^{\left(t\right)}\right)\;⊗\;U^{\left(x\right)}\left(1,R^{\left(t\right)}\right)\dot{+}Q^{\left(3\right)}\left(M,R^{\left(t\right)}\right)\;⊗\;U^{\left(x\right)}\left(2,R^{\left(t\right)}\right)$;  7:$C^{\left(2\right)}\left(2,R^{\left(t\right)}\right)\;=\;C^{\left(1\right)}\left(2,R^{\left(t\right)}\right)\dot{+}Q^{\left(3\right)}\left(2,R^{\left(t\right)}\right)\;⊗\;C^{\left(2\right)}\left(1,R^{\left(t\right)}\right)\dot{+}Q^{\left(3\right)}\left(1,R^{\left(t\right)}\right)\;⊗\;U^{\left(x\right)}\left(3,R^{\left(t\right)}\right)$;  8:**for**  i=3 to *M* **do**  9:     $\;\;\;\;C^{\left(2\right)}\;\left(i,R^{\left(t\right)}\right)\;=\;C^{\left(1\right)}\;\left(i,R^{\left(t\right)}\;\right)\dot{+}Q^{\left(3\right)}\;\left(i,R^{\left(t\right)}\;\right)\;⊗\;C^{\left(2\right)}\;\left(i\;-\;1,R^{\left(t\right)}\;\right)\dot{+}Q^{\left(3\right)}\;\left(i\;-\;1,R^{\left(t\right)}\;\right)\;⊗\;C^{\left(2\right)}\;\left(i\;-\;2,R^{\left(t\right)}\;\right)$; 10:**end for** 11:$R^{\left(t\right)}=\left(r^{\left(p\right)}+1\right):N$; 12:$C^{\left(2\right)}\left(1,R^{\left(t\right)}\right)=C^{\left(1\right)}\left(1,R^{\left(t\right)}\right)⊕Q^{\left(3\right)}\left(1,R^{\left(t\right)}\right)⊕C^{\left(1\right)}\left(M,R^{\left(t\right)}\right)$; 13:**for**  i=2 to *M* **do** 14:   $C^{\left(2\right)}\left(i,R^{\left(t\right)}\right)=C^{\left(1\right)}\left(i,R^{\left(t\right)}\right)⊕Q^{\left(3\right)}\left(i,R^{\left(t\right)}\right)⊕C^{\left(2\right)}\left(i-1,R^{\left(t\right)}\right)$; 15:**end for****Output:** The intermediate ciphertext image $C^{\left(2\right)}$.

### 3.4. Dynamic Binary Scrambling

Similar to dynamic binary diffusion, dynamic binary scrambling enhances the plaintext relevance of CIES-DVEM by incorporating a dynamic nature, thereby overcoming the weaknesses in some existing schemes. As we have already described the row diffusion version of dynamic binary diffusion in the previous subsection, we will introduce the column scrambling version of dynamic binary scrambling here and omit the highly similar row scrambling version. Figure 9 provides a concise example of dynamic binary scrambling.

As can be observed, the first step is to dynamically divide the input image into two partitions. Next, the two partitions are exchanged. Finally, column scrambling is performed on the top and bottom partitions, respectively, so as to obtain the output image of this encryption step. The pseudocode for dynamic binary scrambling is presented in Algorithm 3.

Note that our proposed CIES-DVEM is structurally symmetric. Thus, the decryption process of CIES-DVEM is the inverse of its encryption one. To maintain conciseness, a repetitive description of this inverse process is omitted here.
**Algorithm 3** Dynamic binary scrambling algorithm**Input:** The input image $C^{\left(2\right)}$, $v^{\left(x\right)}$ obtained in hash value stacking, and the key streams $Q^{\left(3\right)}\left(1,3\right)$ and $Q^{\left(5\right)}\left(1:2\times N\right)$.  1:Get the height *M* and width *N* of $C^{\left(2\right)}$;  2:Set $r^{\left(p\right)}=\left(\left(v^{\left(x\right)}+Q^{\left(3\right)}\left(1,2\right)\right)\;\mathrm{mod}\;\left\lfloor M/8\right\rfloor \right)+\left\lfloor M/2\right\rfloor$;  3:Set $C^{\left(3\right)}\in N^{M\times N}$;  4:$C^{\left(3\right)}\left(1:r^{\left(p\right)},:\right)=C^{\left(2\right)}\left(M-r^{\left(p\right)}+1:M,:\right)$  5:$C^{\left(3\right)}\left(r^{\left(p\right)}+1:M,:\right)=C^{\left(2\right)}\left(1:M-r^{\left(p\right)},:\right)$  6:Set $C^{\left(t\right)}=C^{\left(3\right)}\left(1:r^{\left(p\right)},:\right)$;  7:Sort $Q^{\left(5\right)}\left(1:N\right)$ to get the index vector $R^{\left(x\right)}$.  8:**for**  i=1 to *N* **do**  9:   $C^{\left(3\right)}\left(1:r^{\left(p\right)},R^{\left(x\right)}\left(i\right)\right)=C^{\left(t\right)}\left(:,i\right)$; 10:**end for** 11:Set $C^{\left(t\right)}=C^{\left(3\right)}\left(r^{\left(p\right)}+1:M,:\right)$; 12:Sort $Q^{\left(5\right)}\left(N+1:2\times N\right)$ to get the index vector $R^{\left(x\right)}$. 13:**for**  i=1 to *N* **do** 14:   $C^{\left(3\right)}\left(r^{\left(p\right)}+1:M,R^{\left(x\right)}\left(i\right)\right)=C^{\left(t\right)}\left(:,i\right)$; 15:**end for****Output:** The output image $C^{\left(3\right)}$.

## 4. Simulation Experiments

To confirm and emphasize the advantages of CIES-DVEM over some recent advanced encryption schemes, we conducted numerous simulation experiments across eight categories. These experiments include visual effect experiments, key analysis experiments, plaintext sensitivity analysis experiments, and other simulation experiments aimed at evaluating security and efficiency. During the completion of these experiments, we utilized the experimental platform MATLAB R2017a along with hardware components such as the Xeon processor E3-1231 v3 and 8 GB of memory. Moreover, the experimental images utilized were chosen from The USC-SIPI Image Database.

### 4.1. Visual Effect

To be considered a viable image encryption scheme, it is imperative that the scheme has the capability to convert natural images with diverse styles into unidentifiable images that resemble random noise. This means that no useful information can be extracted from the encrypted image without the correct key. To exhibit the encryption and decryption effects of CIES-DVEM, four grayscale and color images with various styles were chosen. Figure 10 depicts the outcomes of our experiments. As one can see, the discernible patterns in natural images are entirely eradicated after CIES-DVEM conducts its encryption processing. However, as long as the correct key is provided, CIES-DVEM is able to convert these unidentifiable encrypted images back to their original plaintext forms with all visual details intact. Accordingly, from a visual perception perspective, CIES-DVEM can effectively protect plaintext images.

### 4.2. Key Space Analysis

Cryptosystems can be subjected to diverse forms of attack; however, the most prevalent type of attack is brute-force attack. Possessing a key space that is sufficiently large, typically no smaller than $2^{128}$, is a crucial feature for competent cryptosystems to effectively counter brute-force attacks [43,44]. In our proposed CIES-DVEM, the secret key comprises four components, which are $x_{0}$, $y_{0}$, *a*, and *b*, as explained in Section 1. Assuming that the computer’s effective representation precision is $10^{-15}$, then the size of CIES-DVEM’s key space is
(11)$$S^{\left(k\right)}=S^{\left(x_{0}\right)}\times S^{\left(y_{0}\right)}\times S^{\left(a\right)}\times S^{\left(b\right)}=8.1\times 10^{61}\approx 2^{205}.$$
Obviously, $S^{\left(k\right)}$ is far greater than $2^{128}$. Hence, CIES-DVEM’s key space is large enough to effectively resist brute-force attacks.

### 4.3. Key Sensitivity Analysis

As per Shannon’s suggestion, the correlation between the secret key and the ciphertext should be highly intricate in terms of statistics [43,44]. Accordingly, any encryption scheme that is competent should feature an exceedingly high level of sensitivity to its secret key. This means that even if the secret key undergoes minimal changes, the resulting ciphertext image should also vary significantly as a result. To examine the key sensitivity of CIES-DVEM, we first generated a random secret key $\hat{K}=\left\{{\hat{x}}_{0},{\hat{y}}_{0},\hat{a},\hat{b}\right\}=\left\{0.367112131687560,\;0.175914170315092,\;0.409479374333251,\;0.912336229763019\right\}$. Then, each component of this secret key was minimally modified. Ultimately, these modified secret keys were employed to encrypt the same experimental image, and the difference between each encrypted image and the original one was calculated. The final experimental results can be found in Figure 11. As one can see, a minor alteration in any of the key components results in a completely distinct encrypted image. This indicates that our proposed CIES-DVEM features extremely high key sensitivity and can satisfy the confusion requirement for robust cryptosystems proposed by Shannon.

### 4.4. Plaintext Sensitivity Analysis

Among various types of attacks, differential attacks are widely considered to be the most threatening. When carrying out differential attacks, attackers often analyze the mathematical relationship between plaintext variations and their corresponding ciphertext variations and then try to break the cryptosystem. Accordingly, a reliable image encryption scheme should exhibit a high level of sensitivity to minor modifications in the plaintext image. To assess the sensitivity of CIES-DVEM when there is a single minimal change in a plaintext image, we modified two pixel bits in 5.1.09. In the first round of modification, the lowest bit of the pixel located in (1,1) was modified. Similarly, in the second round of modification, the lowest bit of the pixel located in (256,256) was modified. The relevant experimental findings are shown in Figure 12. As can be observed, the two difference images between two modified images and 5.1.09 are almost all-zero images, since they differ from 5.1.09 by just one pixel bit each. Notably, even though each modified image only differs by one pixel bit, the encrypted images that correspond to the modified image present fairly significant variations, and the difference images obtained are exceedingly similar to noise images.

Moreover, we also conducted quantitative analyses of CIES-DVEM’s plaintext sensitivity to enable a more objective and accurate assessment of its performance. In our experiments, two popular indicators were introduced to measure the variations in images. One of these indicators is the number of pixels change rate (NPCR), which can reflect the change rate of pixels in the form of a percentage. The following is the mathematical definition of NPCR between the images $M_{1}$ and $M_{2}$:(12)$$NPCR\left(M_{1},M_{2}\right)=\sum_{a=1}^{H}\sum_{b=1}^{W}D\left(a,b\right)/\left(H\times W\right)\times 100\%,$$
where *H* represents the height of these images, *W* denotes the width of these images, and D(a,b) is the difference between $M_{1}\left(a,b\right)$ and $M_{2}\left(a,b\right)$. If $M_{1}\left(a,b\right)=M_{2}\left(a,b\right)$, D(a,b)=0; otherwise, D(a,b)=1. Another indicator that we adopted is the unified average changing intensity (UACI). Similar to NPCR, UACI provides the average intensity of pixel changes as a percentage, which can be calculated as follows:(13)$$UACI\left(M_{1},M_{2}\right)=\sum_{a=1}^{H}\sum_{b=1}^{W}\frac{\left|M_{1}\left(a,b\right)-M_{2}\left(a,b\right)\right|}{255\times H\times W}\times 100\%.$$
We employed 10 experimental images to quantitatively analyze the plaintext sensitivity of CIES-DVEM. The experimental results obtained are listed in Table 2 and Table 3. As one can see, in all six encryption schemes, the two average values (99.6095 and 33.4633) obtained by CIES-DVEM are the closest to the ideal values (99.6094 and 33.4635) and also exhibit the highest level of stability (0.0087 and 0.0406). Hence, CIES-DVEM does possess outstanding plaintext sensitivity and can effectively withstand different types of differential attacks.

### 4.5. Histogram Analysis

Regarding pixel distribution, plaintext images typically exhibit notable features. Undoubtedly, to prevent information leakage, it is important for an encryption scheme to maintain a uniform distribution of ciphertext pixels. In our experiments, histograms are utilized to visually depict the distributions of pixels in images. The relevant histogram analysis results are illustrated in Figure 13. Upon observation, it can be discerned that the distributions of pixels in natural images are remarkably uneven, and the features within them are highly distinct. However, following the encryption transformation of CIES-DVEM, the pixel distributions become highly uniform, leaving no notable features. Thus, CIES-DVEM has the capability to effectively defend against different attacks that rely on pixel distribution.

### 4.6. Correlation Analysis

Adjacent pixels in plaintext images exhibit noteworthy correlations. Hence, a suggested encryption scheme should entirely eradicate such correlations. The ability of CIES-DVEM to eliminate the correlations is depicted in Figure 14. By observing Figure 14, it can be clearly seen that in three plaintext images, adjacent pixels in three (Horizontal, Vertical, and Diagonal) directions have strong correlations close to 1. However, in the three ciphertext images constructed by CIES-DVEM, such strong correlations no longer exist. The pixel correlations in all directions are extremely close to 0. In other words, CIES-DVEM does have exceptional performance in eliminating strong correlations between adjacent pixels.

To further assert the superiority of CIES-DVEM, we performed supplementary quantitative analyses by introducing the correlation coefficient (CC). Mathematically, one can employ the following definition to calculate CC:(14)$$CC=\frac{E(\left(V_{a}-E\left(V_{a}\right)\right)\left(V_{b}-E\left(V_{b}\right)\right))}{\sqrt{D\left(V_{a}\right)D\left(V_{b}\right)}}.$$
In Equation (Equation 14), E(V) denotes the expectation of *V*, whereas D(V) stands for the variance of *V*. In addition, $V_{a}$ and $V_{b}$ represent the values of two adjacent pixels. After conducting numerous experiments, we finally obtained the results that are listed in Table 4. Observing the results, it can be inferred that the quantitative analysis results are entirely consistent with the findings of the graphical analysis. All of the images exhibit extremely high CC values prior to encryption. However, following encryption, the CC values across all directions are significantly reduced to almost negligible levels close to 0.

### 4.7. Information Entropy Analysis

Information entropy can serve as a representation for the distribution and randomness of signal sources, which in turn can assist in verifying the encryption schemes’ security. For encrypted images with an 8-bit depth, having an entropy value of 8 would be the most optimal. In mathematical terms, one can calculate information entropy through
(15)$$I\left(\psi \right)=-\sum_{k=1}^{M}\rho (\psi_{k})\log_{2}\rho (\psi_{k}).$$
In Equation (Equation 15), the quantity of symbols ψ is *M*, while $\rho (\psi_{k})$ represents the probability of $\psi_{k}$. As for an encrypted image, a higher entropy value indicates that the randomness of its pixels is greater and the distribution of the pixels is more even. Table 5 lists the entropy values of ten frequently utilized experimental images and their corresponding encrypted counterparts. It can be observed that the entropy values of all plaintext images are comparatively low. Nevertheless, the ciphertext images created by CIES-DVEM exhibit entropy values that are in close proximity to the optimal value of 8. Hence, the ciphertext images produced by CIES-DVEM are characterized by outstanding randomness and uniform distribution.

Moreover, we also conducted comparative experiments. The pertinent findings from our experiments are provided in Table 6. As we can see, each scheme has attained an entropy value that is almost equal to 8. This suggests that these schemes demonstrate strong performance regarding randomness and distribution uniformity. Significantly, CIES-DVEM has achieved the highest entropy value, thereby further proving its superiority.

### 4.8. Robustness Analysis

While transmitting ciphertext images, it is likely that some data may become lost or corrupted, and if a malicious attack occurs, these ciphertext images may also undergo similar damage [3]. Accordingly, for the purpose of investigating the robustness of CIES-DVEM, we intentionally processed the encrypted image produced by CIES-DVEM. In Figure 15, the experimental outcomes for four encrypted images losing approximately 14% to 50% of their pixels are shown in the first two rows. Upon observation, we can discover that the loss of a large amount of data makes the decrypted image blurry, but it does not prevent us from perceiving the majority of visual information from it. Similarly, when we intentionally add varying intensities of salt and pepper noise (0.03/0.06/0.09/0.12) to the encrypted image, CIES-DVEM is also able to effectively reconstruct most of the patterns contained in the plaintext image. Thus, one can derive the conclusion that CIES-DVEM boasts excellent robustness and can efficiently withstand malicious attacks that result in data loss or corruption.

### 4.9. Efficiency Analysis

Undoubtedly, a qualified encryption scheme must possess exceptionally high encryption efficiency while ensuring security; otherwise, it would be unsuitable for today’s high-throughput application environments. Accordingly, we implemented several targeted measures in the design of CIES-DVEM to guarantee its encryption efficiency. Firstly, we optimized the method of utilizing the hash value, allowing for the chaotic sequences to be created beforehand and eliminating the need to constantly replace chaotic sequences. Secondly, we boosted the efficiency of using chaotic sequences. To encrypt an image of size M×N, CIES-DVEM only requires the employing of sequences with a total length of M+N+2×M×N, which is superior to many recent encryption schemes. Finally, we introduced many vector-level operations in the encryption process instead of pixel-level or bit-level operations commonly adopted in other encryption schemes, which further optimizes the encryption efficiency significantly. Table 7 presents the times and throughputs achieved by CIES-DVEM and five other recent schemes. As one can observe, for images of each common size, CIES-DVEM requires the shortest average encryption time and attains the highest average encryption speed.

## 5. Conclusions

In this study, to tackle the flaws found in certain advanced image encryption schemes, we first established a strong 2D hyperchaotic map called 2D-ELMM. With the help of a series of chaos evaluation metrics, including LE, SE, and KE, the superiority of 2D-ELMM was confirmed. Related experiments and analyses indicate that 2D-ELMM possesses a simple structure, a wide range of hyperchaos, a uniform trajectory distribution, a fast trajectory divergence rate, and excellent chaotic performance. Consequently, it is highly suitable for image encryption.

Moreover, by exploiting 2D-ELMM and dynamic vector-level operations, we further devised a novel and efficient image encryption scheme named CIES-DVEM. This suggested CIES-DVEM consists of eight encryption steps, which are the generation of key streams, hash value stacking, two rounds of dynamic binary diffusion, and four rounds of dynamic binary scrambling. In CIES-DVEM, the first encryption step generates a chaotic sequence that corresponds to the secret key and converts it into the key streams needed for the following encryption steps. Hash value stacking takes the hash value of the input image and the key streams to generate two matrices. These matrices are then stacked onto the input image. Dynamic binary diffusion and dynamic binary scrambling introduce plaintext-related parameters to dynamically divide the intermediate ciphertext image into two partitions and then diffuse and scramble them in different ways. Note that unlike some existing algorithms, both the diffusion operations and the scrambling operations adopted in CIES-DVEM are dynamic depending on the plaintext. Therefore, CIES-DVEM has excellent plaintext sensitivity and can effectively resist various plaintext attacks. Moreover, all encryption steps in CIES-DVEM are not pixel- or bit-level but vector-level, so CIES-DVEM achieves superior encryption efficiency beyond most existing encryption schemes. As demonstrated by numerous experiments and analyses carried out afterwards, CIES-DVEM not only has great advantages in terms of encryption efficiency, but it also outperforms many recent schemes in practicality and security.

In the future, we will continue to enhance and optimize the proposed CIES-DVEM. For instance, a specific encryption step may be introduced to acquire plaintext features instead of relying on the SHA-256 hash function. Furthermore, our future research will try to introduce techniques such as compressed sensing, regions of interest, and neural networks.

## Figures and Tables

**Figure 1 entropy-25-01147-f001:**
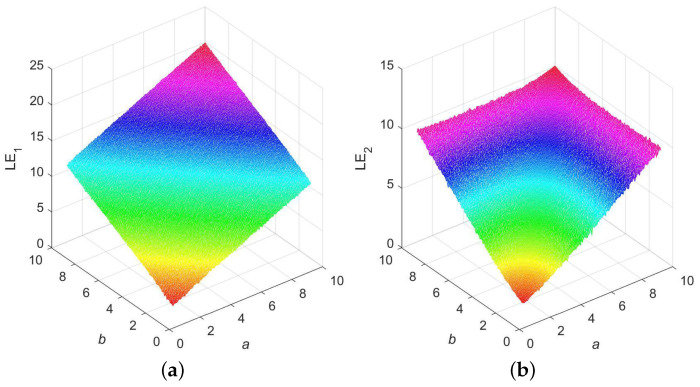
LE presentations for 2D-ELMM: (**a**) $LE_{1}$; (**b**) $LE_{2}$.

**Figure 2 entropy-25-01147-f002:**
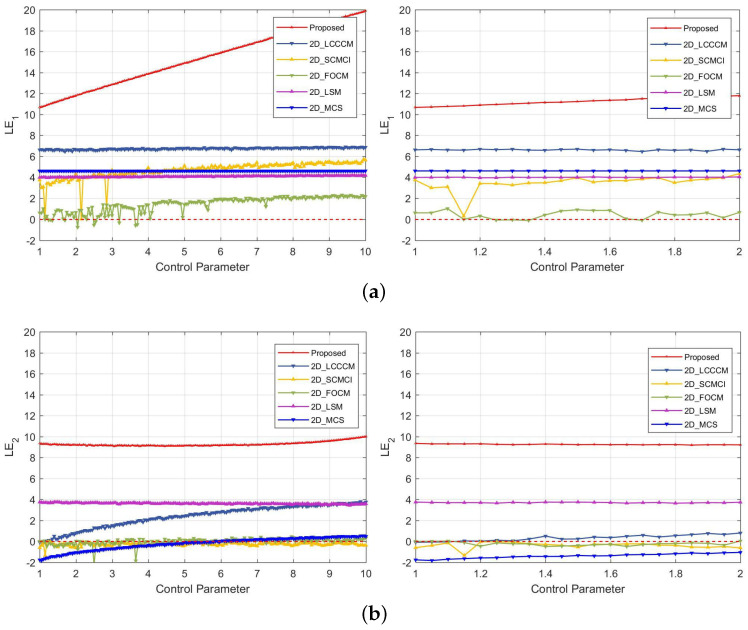
LE comparison for 2D-ELMM and five other newly proposed 2D maps: (**a**) $LE_{1}$; (**b**) $LE_{2}$.

**Figure 3 entropy-25-01147-f003:**
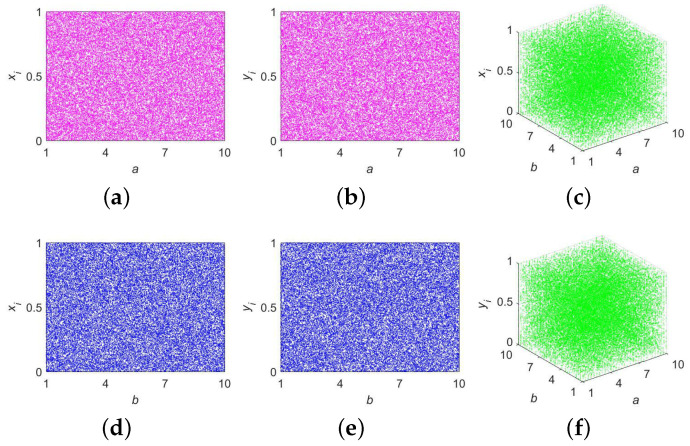
Bifurcation diagrams of 2D-ELMM: (**a**) 2D diagram for $x_{i}$ when b=10; (**b**) 2D diagram for $y_{i}$ when b=10; (**c**) 3D diagram for $x_{i}$; (**d**) 2D diagram for $x_{i}$ when a=10; (**e**) 2D diagram for $y_{i}$ when a=10; (**f**) 3D diagram for $y_{i}$.

**Figure 4 entropy-25-01147-f004:**
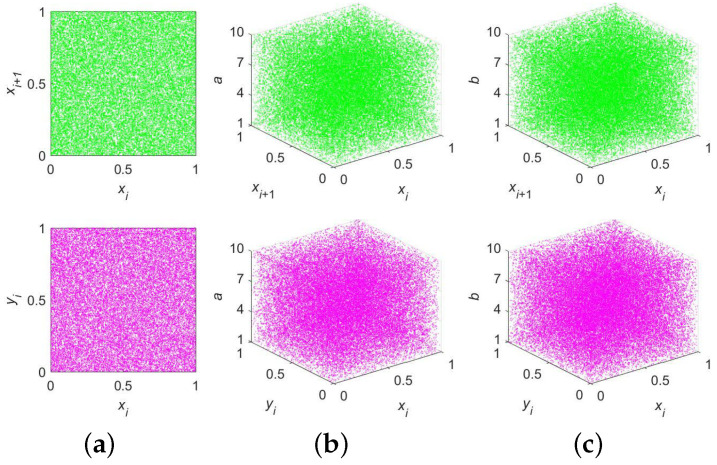
Trajectory diagrams of 2D-ELMM: (**a**) (a,b)=(10,10); (**b**) b=10; (**c**) a=10.

**Figure 5 entropy-25-01147-f005:**
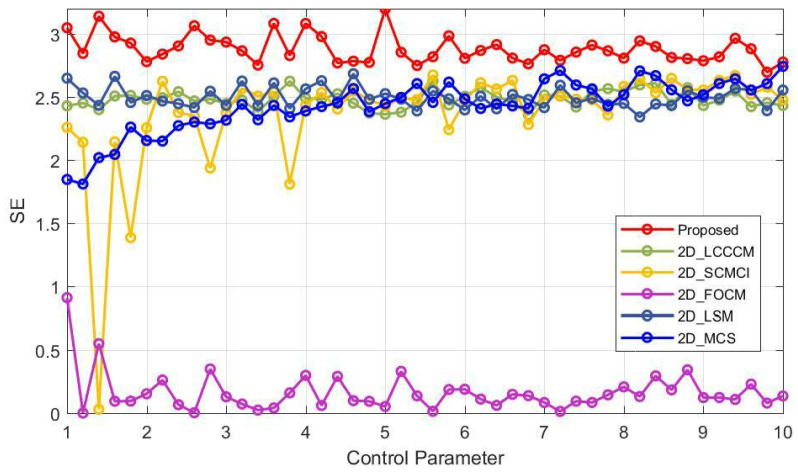
SE comparison for 2D-ELMM and five other newly proposed 2D maps.

**Figure 6 entropy-25-01147-f006:**
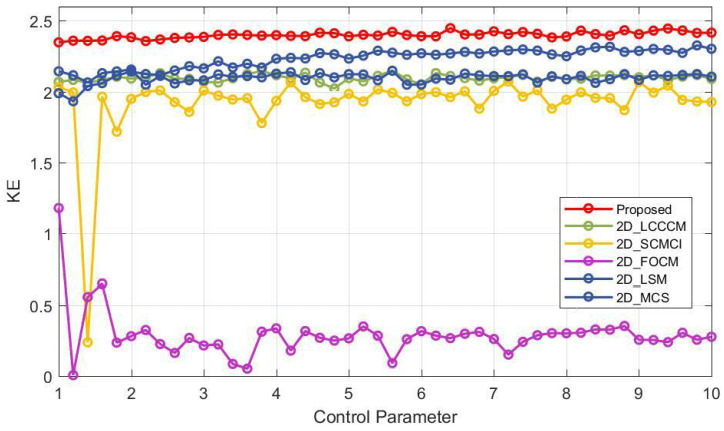
KE comparison for 2D-ELMM and five other newly proposed 2D maps.

**Figure 7 entropy-25-01147-f007:**
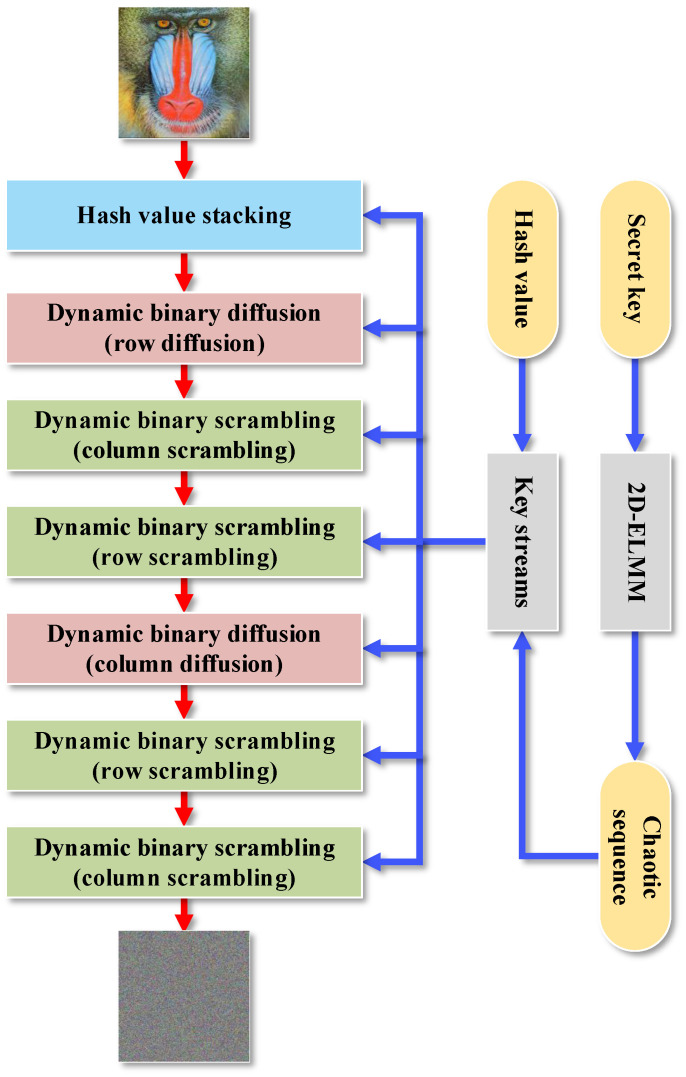
Encryption flowchart for CIES-DVEM.

**Figure 8 entropy-25-01147-f008:**
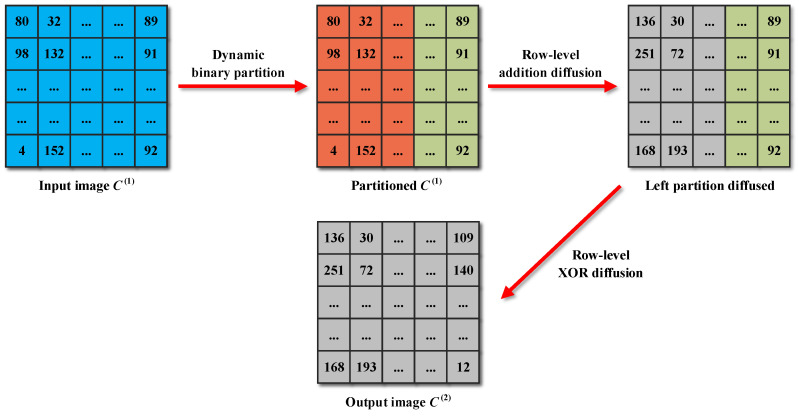
Flowchat of dynamic binary diffusion.

**Figure 9 entropy-25-01147-f009:**
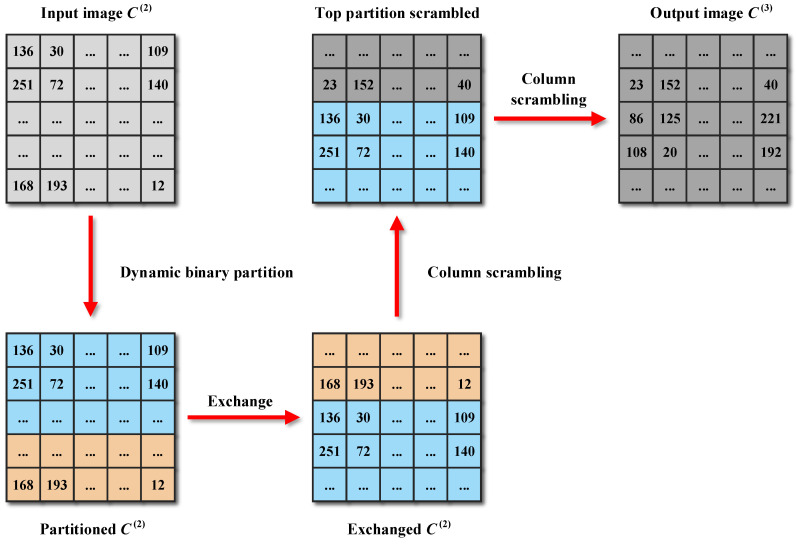
Flowchat of dynamic binary scrambling.

**Figure 10 entropy-25-01147-f010:**
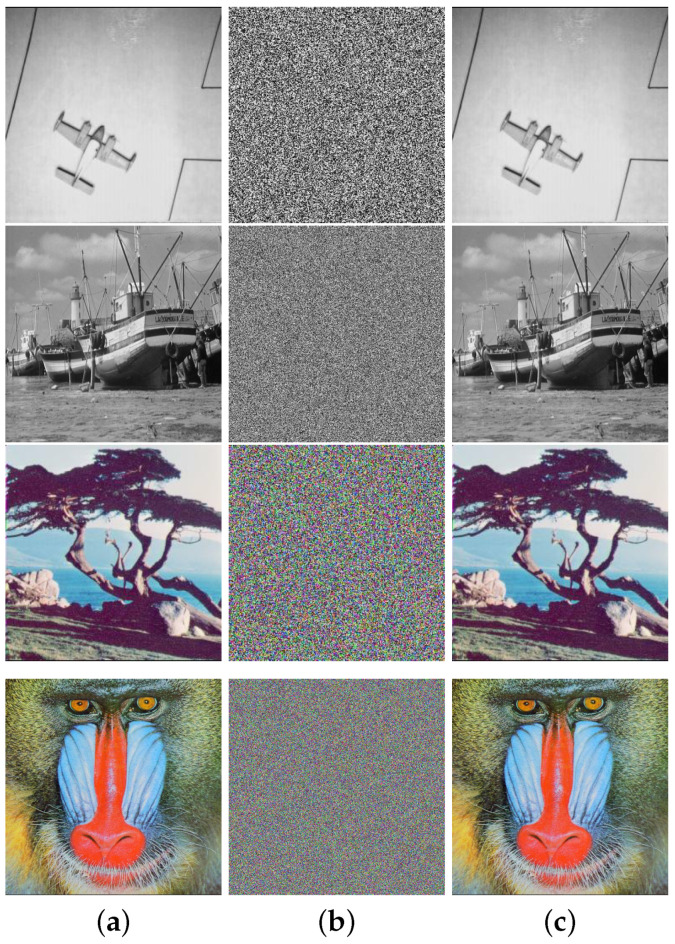
Encryption and decryption effects of CIES-DVEM: (**a**) plaintext images; (**b**) encrypted images; (**c**) decrypted images.

**Figure 11 entropy-25-01147-f011:**
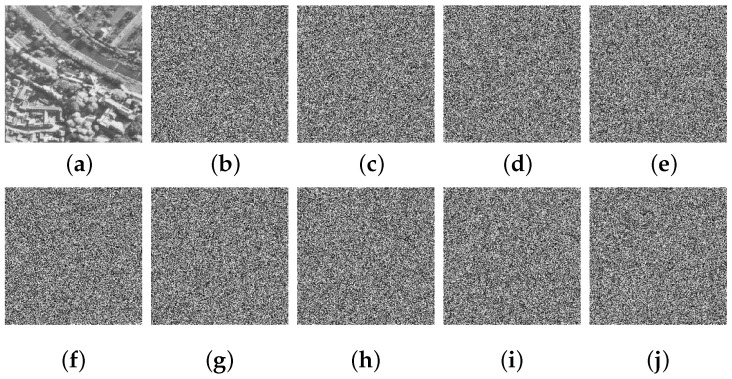
Key sensitivity experimental results: (**a**) 5.1.14; (**b**) ciphertext obtained by ${\hat{x}}_{0}={\hat{x}}_{0}+10^{-15}$; (**c**) ${\hat{y}}_{0}={\hat{y}}_{0}+10^{-15}$; (**d**) $\hat{a}=\hat{a}+10^{-15}$; (**e**) $\hat{b}=\hat{b}+10^{-15}$; (**f**) ciphertext of (**a**); (**g**) difference between (**f**) and (**b**); (**h**) difference between (**f**) and (**c**); (**i**) difference between (**f**) and (**d**); (**j**) difference between (**f**) and (**e**).

**Figure 12 entropy-25-01147-f012:**
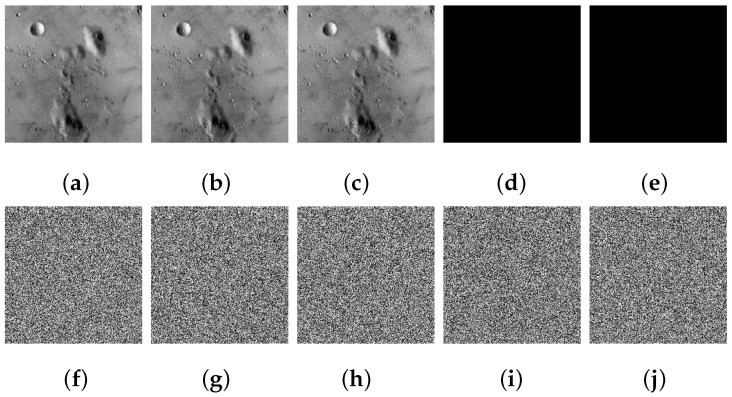
Plaintext sensitivity experimental results: (**a**) 5.1.09; (**b**) one pixel bit modified at (1,1); (**c**) one pixel bit modified at (256,256); (**d**) difference between (**a**) and (**b**); (**e**) difference between (**a**) and (**c**); (**f**) ciphertext of (**a**); (**g**) ciphertext of (**b**); (**h**) ciphertext of (**c**); (**i**) difference between (**f**) and (**g**); (**j**) difference between (**f**) and (**h**).

**Figure 13 entropy-25-01147-f013:**
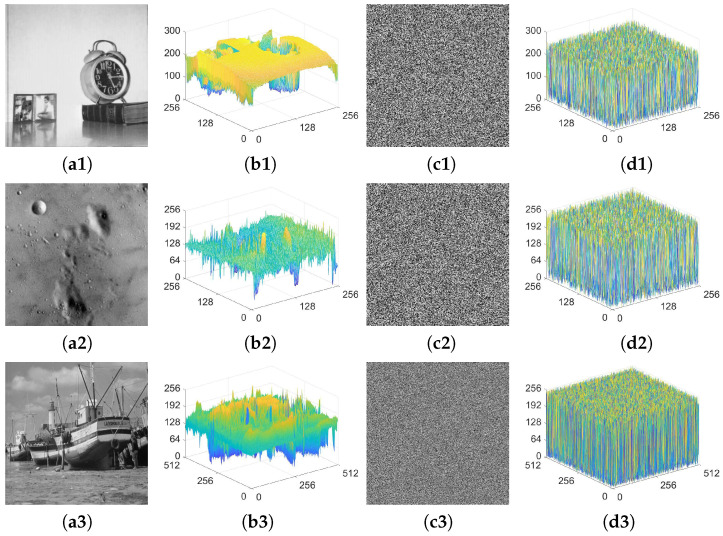
Histogram analysis results for CIES-DVEM: (**a1**) 5.1.12; (**b1**) histogram for (**a1**); (**c1**) ciphertext of (**a1**); (**d1**) histogram for (**c1**); (**a2**) 5.1.09; (**b2**) histogram for (**a2**); (**c2**) ciphertext of (**a2**); (**d2**) histogram for (**c2**); (**a3**) boat.512; (**b3**) histogram for (**a3**); (**c3**) ciphertext of (**a3**); (**d3**) histogram for (**c3**).

**Figure 14 entropy-25-01147-f014:**
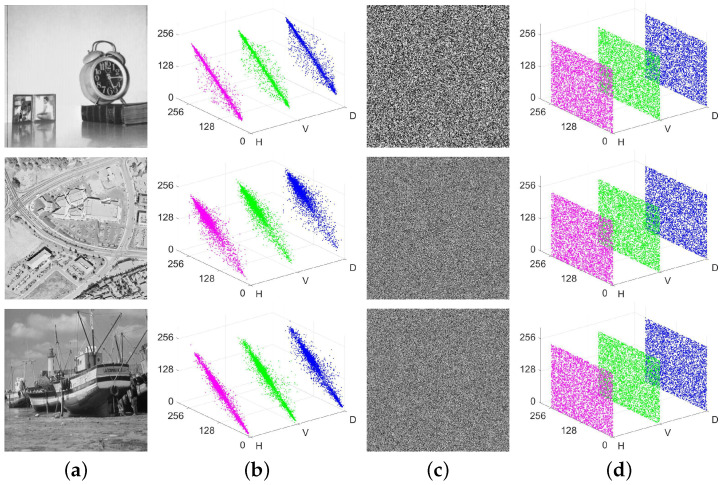
Visual representations of pixel correlations for plaintext and encrypted images: (**a**) three plaintext images; (**b**) correlation representations of (**a**); (**c**) encrypted images of (**a**); (**d**) correlation representations of (**c**).

**Figure 15 entropy-25-01147-f015:**
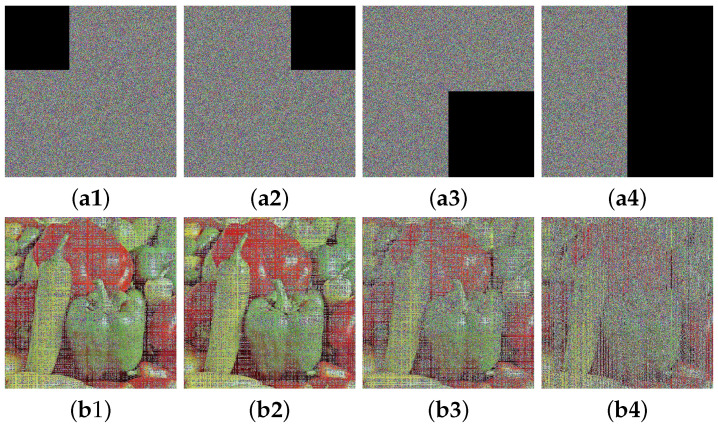
Robustness analysis results for CIES-DVEM: (**a1**–**a4**) are four encrypted images with data missing from different locations; (**b1**–**b4**) are decrypted images for (**a1**–**a4**); (**a5**–**a8**) are four encrypted images with salt and pepper noise intensities of 0.03/0.06/0.09/0.12, respectively; (**b5**–**b8**) are decrypted images for (**a5**–**a8**).

**Table 1 entropy-25-01147-t001:** Five recently proposed leading 2D chaotic maps.

Reference	Name	Year	Definition	Ctrl. Parms.
[28]	2D-LCCCM	2022	$\left\{\begin{array}{c} x_{i+1}=\cos\left(\pi^{2}\left(4\mu x_{i}\left(1-x_{i}\right)+py_{i}\left(1-y_{i}^{2}\right)\right)+\pi /2\right)\\ y_{i+1}=\cos\left(\pi^{2}\left(4\mu y_{i}\left(1-y_{i}\right)+px_{i}\left(1-x_{i+1}^{2}\right)\right)+\pi /2\right) \end{array}\right.$	μ,p
[29]	2D-SCMCI	2021	$\left\{\begin{array}{c} x_{i+1}=r\sin\left(\pi \left(\left(y_{i}+h\right)k\sin\left(a\pi /x_{i}\right)\right)\right)\\ y_{i+1}=r\sin\left(kx_{i+1}+h\right)\sin\left(a\pi /x_{i}\right))) \end{array}\right.$	r,h,k,a
[21]	2D-FOCM	2022	$\left\{\begin{array}{c} x_{i+1}=x_{i}+\left(h^{v}/\left(\Gamma \left(1+v\right)\right)\right)\cos\left(2\pi x_{i}/\left(2\mu x_{i}^{4}-1\right)-y_{i}\right)\\ y_{i+1}=y_{i}+\left(h^{v}/\left(\Gamma \left(1+v\right)\right)\right)\cos\left(\mu \pi x_{i+1}+y_{i}\right) \end{array}\right.$	h,v,μ
[16]	2D-LSM	2021	$\left\{\begin{array}{c} x_{i+1}=\cos\left(4ax_{i}\left(1-x_{i}\right)+b\sin\left(\pi y_{i}\right)+1\right)\\ y_{i+1}=\cos\left(4ay_{i}\left(1-y_{i}\right)+b\sin\left(\pi x_{i}\right)+1\right) \end{array}\right.$	a,b
[23]	2D-MCS	2020	$\left\{\begin{array}{c} x_{i+1}=-ax_{i}/\left(1+y_{i}^{2}\right)\;\mathrm{mod}\;N\\ y_{i+1}=\left(x_{i}+by_{i}\right)\;\mathrm{mod}\;N \end{array}\right.$	a,b,N

**Table 2 entropy-25-01147-t002:** NPCR results of CIES-DVEM and other scheme (%).

Size	Filename	CIES-DVEM	[17]	[18]	[20]	[16]	[19]
256×256	5.1.10	99.6124	99.5743	99.5850	99.6094	99.6002	99.6201
	5.1.11	99.5956	99.6323	99.5956	99.6189	99.6023	99.5926
	5.1.12	99.6108	99.6424	99.5758	99.6178	99.5809	99.5941
	5.1.13	99.6109	99.5972	99.6048	99.5956	99.5926	99.5987
	5.1.14	99.6078	99.6429	99.6124	99.6075	99.6165	99.5895
512×512	5.2.09	99.6170	99.5941	99.5956	99.5850	99.5928	99.6124
	7.1.01	99.6048	99.6353	99.5953	99.6006	99.6181	99.6067
	7.1.02	99.6246	99.6002	99.6002	99.6170	99.6040	99.6007
	7.1.03	99.5972	99.6025	99.6220	99.6281	99.6054	99.6021
	Boat.512	99.6140	99.6128	99.6197	99.6178	99.6152	99.6133
	Average	99.6095	99.6134	99.6006	99.6098	99.6028	99.6030
	Std. Dev.	0.0087	0.0236	0.0146	0.0129	0.0119	0.0100

**Table 3 entropy-25-01147-t003:** UACI results of CIES-DVEM and other scheme (%).

Size	Filename	CIES-DVEM	[17]	[18]	[20]	[16]	[19]
256×256	5.1.10	33.4835	33.4260	33.4917	33.4801	33.3818	33.4701
	5.1.11	33.3793	33.4607	33.3859	33.5077	33.4919	33.4787
	5.1.12	33.4883	33.4216	33.3798	33.4835	33.4507	33.4670
	5.1.13	33.4421	33.4792	33.5620	33.5054	33.3901	33.4727
	5.1.14	33.4086	33.4775	33.6904	33.4667	33.4541	33.5750
512×512	5.2.09	33.4727	33.4426	33.4174	33.4687	33.5047	33.4473
	7.1.01	33.5037	33.4934	33.4885	33.4514	33.4971	33.5300
	7.1.02	33.4961	33.3665	33.4248	33.4628	33.5180	33.3661
	7.1.03	33.4773	33.5255	33.4990	33.5650	33.4791	33.5262
	Boat.512	33.4814	33.4722	33.5334	33.4590	33.4627	33.4727
	Average	33.4633	33.4565	33.4873	33.4850	33.4630	33.4806
	Std. Dev.	0.0406	0.0445	0.0941	0.0338	0.0461	0.0559

**Table 4 entropy-25-01147-t004:** Quantitative analysis results by employing CC.

Size	Filename	Plaintext				Ciphertext		
		H	V	D		H	V	D
256×256	5.1.10	0.8679	0.9095	0.8161		0.0008	0.0020	−0.0008
	5.1.12	0.9709	0.9558	0.9397		−0.0024	0.0007	0.0018
	5.1.14	0.9709	0.9558	0.9397		−0.0013	0.0003	0.0014
512×512	5.2.09	0.8622	0.9037	0.8024		−0.0021	0.0003	−0.0010
	7.1.01	0.9217	0.9640	0.9063		−0.0022	−0.0009	0.0013
	boat.512	0.9679	0.9379	0.9268		0.0023	0.0019	−0.0024
1024×1024	5.3.02	0.9003	0.9158	0.8476		0.0020	−0.0019	0.0017
	7.2.01	0.9505	0.9630	0.9435		0.0015	0.0009	−0.0018
	testpat.1k	0.8242	0.7458	0.7146		−0.0009	0.0009	−0.0012

**Table 5 entropy-25-01147-t005:** Information entropy experimental outcomes for CIES-DVEM.

Size	Filename	Plaintext	Ciphertext
512×512	5.2.08	7.2010	7.9994
	5.2.10	5.7056	7.9992
	7.1.02	4.0045	7.9994
	7.1.04	6.1074	7.9994
	7.1.06	6.6953	7.9993
	7.1.08	5.0534	7.9994
	ruler.512	0.5000	7.9993
1024×1024	5.3.01	7.5237	7.9998
	5.3.02	6.8303	7.9999
	7.2.01	5.6415	7.9998

**Table 6 entropy-25-01147-t006:** Entropy values of different Lena ciphertext images.

Scheme	Inf. Entropy
CIES-DVEM	7.9994
[17]	7.9993
[18]	7.9976
[20]	7.9993
[16]	7.9992
[19]	7.9992

**Table 7 entropy-25-01147-t007:** Times (throughputs) achieved by CIES-DVEM and five recent schemes.

Size	256×256	512×512	1024×1024	Average
CIES-DVEM	0.0203 s	0.0878 s	0.3755 s	–
	(24.6305 Mbps)	(22.7790 Mbps)	(21.3049 Mbps)	(22.9048 Mbps)
[17]	0.0768 s	0.3213 s	1.3806 s	–
	(6.5104 Mbps)	(6.2247 Mbps)	(5.7946 Mbps)	(6.1766 Mbps)
[18]	0.1524 s	0.6313 s	2.5712 s	–
	(3.2808 Mbps)	(3.1681 Mbps)	(3.1114 Mbps)	(3.1868 Mbps)
[45]	0.0915 s	0.4088 s	2.0314 s	–
	(5.4645 Mbps)	(4.8924 Mbps)	(3.9382 Mbps)	(4.7650 Mbps)
[19]	0.4341 s	1.7586 s	7.1223 s	–
	(1.1518 Mbps)	(1.1373 Mbps)	(1.1232 Mbps)	(1.1374 Mbps)
[46]	0.0800 s	0.4842 s	2.2848 s	–
	(6.2500 Mbps)	(4.1305 Mbps)	(3.5014 Mbps)	(4.6273 Mbps)

## Data Availability

Not applicable.

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
