# Peer review of "Exploiting Dynamic Vector-Level Operations and a 2D-Enhanced Logistic Modular Map for Efficient Chaotic Image Encryption"

_entropy, 2023, doi:10.3390/e25081147_

Round 1
Reviewer 1 Report
Dear editor and authors
This work considers the problem of image encryption through a multi-level encryption process that utilizes a novel chaotic map.
I found the work overall interesting, i enjoyed reading it, and i believe it holds merit to the community. But nonetheless, i believe there are many parts that require further clarification, and should be described better. Also, the authors should consider reproducability of their work, so with this thought in mind, some portions need to be better desrcibed.
My points are outlined below:
--The vector level operations that is in the title, i believe is not explored enough. The authors write this as a novelty, buti did not particularly saw it as that much of an improvement. I mean, there are many works that consider vector level operations, for example instead of shuffling all the pixels, they shuffle all the rows and columns. This is less ecure, but can be peformed significantly faster. I did not feel this was such a novel part, i believe it has more to do with good and organized code writing. I belive this part at the discussion section at the end must be expanded a bit to clarify that.
--"Classic 1D chaotic maps have weaknesses such as uneven trajectory distributions and 84
discontinuous chaotic intervals, which cannot well meet the needs ..." This statement is inaccurate, as these problems are not limited to 1d maps. A 2d map is equally possible to show these exact problems. To solve these we need to apply different types of nonlinearities, like the modulo operator, not increase dimension.
--"It should be pointed out 87
that chaotic systems have a high dimensionality, typically three or four dimensions, that 88an cause effiiency issues that are unacceptable for many application" again this is inaccurate. Inc ontinuous time there require at least 3 dimensions. But in discrete time we can have chaos in a single dimension.
--THe map proposed is very interesting, but to me, the comparison performed in table 1 feels like 'cheating'. Let me explain this. Although the authors indeed compare their map with many recent ones from the literature, all the maps they chose have indeed too many operations, but note of them uses the modulo. The modulo function is one of the best nonlinearities, they use it themselves, and it basically highly boosts the behavior of the proposed map. So comparing their map with others that do not use modulo is for me very unfair comparison.
There are noumerous maps with a modulo from the literature, the comparison must be performed with them. These maps must also be commented upon in the introduction
--Hua, Z.; Zhang, Y.; Zhou, Y. Two-dimensional modular chaotification system for improving chaos complexity. IEEE Trans. Signal Process. 2020, 68, 1937–1949.
--Lyapunov Exponent Enhancement in Chaotic Maps with Uniform Distribution Modulo One Transformation.
-- Zhang, Z., H. Zhu, P. Ban, Y. Wang, and L. Y. Y. Zhang, 2022 Buffeting chaotifiation model for enhancing chaos and its hardware implementation. IEEE Transactions on Industrial Electronics.
--Hua, Z., Chen, Y., Bao, H., & Zhou, Y. (2021). Two-dimensional parametric polynomial chaotic system. IEEE Transactions on Systems, Man, and Cybernetics: Systems, 52(7), 4402-4414.
--the parameter range comparison for the LEs is a bit unfair. The parameter may range between 0 and 10, but in your map, the terms are exponential so basically we bifurcate between e^0=1 and e^10=22026.4657948, so understandably your map can obtain a much higher LE.
--In figure 4, I would also suggest including phase diagrams for the pairs (x(i-1), x(i)) so that we can see the connection between consecutive values of the states.
--Section 3.1, I do not understand, how we obtain the Q^0 series. this is clearly the values of the chaotic map, but how are the parameters chosen? Are they not plaintext dependent?
--Also, in the same part, why is tha range of the time series M+N+2*M*n? SInce we have a 2d map, we can utilize each state, so the time series can be shorter. Please clarify.
--Algorithm 2 and 3 are to me very unclear. Having the pseudocode is a good idea, but all i am seeing are some operations, many xors, and some sort of shuffling, which I am not sure what they do exactly in the big plan of the process. You need to clearly describe the operations in the text.
--I think this also needs some clarification, you specify the keys of the process, but from what i understand the hash of the image is also used. Is this hash also fed to the map parameters?
--in figure 13, the 3d histogram in my opinion looks pointlessly complex. It would be much better if you simply included a 2d figure a histogram. What do the 3d axes even represent?
--figure 15, try also cutting larger parts of the image, like 50% and 75%.
--"o encrypt an image of size M × N, CIES-DVEM only requires the employing of sequences with a total length of 323
M + N + 2 × M × N, which is superior to many recent encryption schemes". I am not sure if this is really correct. Certainly to perform xor, which every work does, we need M*N*8 bits to be computed, or as you say, M*N byte values. and then if we were to shuffle the rows and columns, we would again need M permutation values and N permutation values. So M*N+M+N is a standardized kind of number. Saying that is it a short number is an exageration. Let's also not forget that you are hashing as well, this again requires some computation.
--"Firstly, we optimized the method of utilizing the hash value, allowing for the chaotic sequences to
be created beforehand and eliminating the need to constantly replace chaotic sequences" SO in your computational times reported, you do not include the time to generate the chaotic sequences? I believe this is unfair and wrong. The time for computing the chaotic sequence of length M*N+M+N must be included. And again i must ask, since the map is 2d, why not half of that length?
--The rest of the simulation tests that are performed are all fine.
Overall, i found some inaccuracies, and not a good enough discreption of the process. Unless these issues are correctly addressed, the work cannot proceed with publication. Please consider reproducability of your results, and a very clear description of what is performed in each step. SHould you feel it is required, even add a simple numerical example of a very small (10x10) matrix to make the process clear.
For all these reasons, I would suggest major revisions for this work.
Author Response
We would like to express our sincere thanks to the reviewers for their constructive comments and suggestions. Following the comments and suggestions, we have carefully revised our manuscript.
Below you will find our point-by-point responses to the reviewers' comments.
* * * * * * * * * * * * * * * * * * * * * * * * * * * * * * * * * * * * * * * * * * * * * *
Response to the comments of Reviewer 1
- This work considers the problem of image encryption through a multi-level encryption process that utilizes a novel chaotic map. I found the work overall interesting, I enjoyed reading it, and I believe it holds merit to the community. But nonetheless, I believe there are many parts that require further clarification, and should be described better. Also, the authors should consider the reproducibility of their work, so with this thought in mind, some portions need to be better described. My points are outlined below:
We would like to express our heartfelt gratitude for your affirmation of our work. Furthermore, for the problems and deficiencies you pointed out, we have also made serious revisions based on your valuable comments and suggestions. The relevant modifications are highlighted in yellow.
- The vector level operations that is in the title, I believe is not explored enough. The authors write this as a novelty, but I did not particularly saw it as that much of an improvement. I mean, there are many works that consider vector level operations, for example instead of shuffling all the pixels, they shuffle all the rows and columns. This is less secure, but can be performed significantly faster. I did not feel this was such a novel part, I believe it has more to do with good and organized code writing. I believe this part at the discussion section at the end must be expanded a bit to clarify that.
We would like to express our sincere thanks for your constructive comments and suggestions on these points.
According to your valuable review comment, we have revised the Conclusions section by adding the following clarification:
Note that unlike some existing algorithms, both the diffusion operations and the scrambling operations adopted in CIES-DVEM are dynamic depending on the plaintext. Therefore, CIES-DVEM has excellent plaintext sensitivity and can effectively resist various plaintext attacks. Moreover, all encryption steps in CIES-DVEM are not pixel- or bit-level but vector-level, so CIES-DVEM achieves superior encryption efficiency beyond most existing encryption schemes.
The relevant modifications are highlighted in yellow.
- "Classic 1D chaotic maps have weaknesses such as uneven trajectory distributions and discontinuous chaotic intervals, which cannot well meet the needs ..." This statement is inaccurate, as these problems are not limited to 1d maps. A 2d map is equally possible to show these exact problems. To solve these, we need to apply different types of nonlinearities, like the modulo operator, not increase dimension.
We would like to express our sincere thanks for your constructive and valuable comment on this point.
According to your valuable review comment, we have revised the original description to be more precise and objective. The specific revisions are as follows:
Classic chaotic maps like the logistic map and tent map have weaknesses such as uneven trajectory distributions and discontinuous chaotic intervals, which cannot well meet the needs of many applications, including image encryption.
The relevant modifications are highlighted in yellow.
- "It should be pointed out that chaotic systems have a high dimensionality, typically three or four dimensions, that can cause efficiency issues that are unacceptable for many application" again this is inaccurate. In continuous time there require at least 3 dimensions. But in discrete time we can have chaos in a single dimension.
We sincerely appreciate your constructive and valuable comment on this point.
According to your valuable review comment, we have revised the original description to be more precise and objective. The specific revisions are as follows:
It should be pointed out that if a chaotic system has too many dimensions, such as three or four, it may cause efficiency problems that are unacceptable for many applications.
The relevant modifications are highlighted in yellow.
- The map proposed is very interesting, but to me, the comparison performed in table 1 feels like 'cheating'. Let me explain this. Although the authors indeed compare their map with many recent ones from the literature, all the maps they chose have indeed too many operations, but note of them uses the modulo. The modulo function is one of the best nonlinearities, they use it themselves, and it basically highly boosts the behavior of the proposed map. So, comparing their map with others that do not use modulo is for me very unfair comparison.
There are numerous maps with a modulo from the literature, the comparison must be performed with them. These maps must also be commented upon in the introduction
--Hua, Z.; Zhang, Y.; Zhou, Y. Two-dimensional modular chaotification system for improving chaos complexity. IEEE Trans. Signal Process. 2020, 68, 1937–1949.
--Lyapunov Exponent Enhancement in Chaotic Maps with Uniform Distribution Modulo One Transformation.
-- Zhang, Z., H. Zhu, P. Ban, Y. Wang, and L. Y. Y. Zhang, 2022 Buffeting chaotifiation model for enhancing chaos and its hardware implementation. IEEE Transactions on Industrial Electronics.
--Hua, Z., Chen, Y., Bao, H., & Zhou, Y. (2021). Two-dimensional parametric polynomial chaotic system. IEEE Transactions on Systems, Man, and Cybernetics: Systems, 52(7), 4402-4414.
We would like to express our sincere thanks for your constructive comments and suggestions on this point.
According to your valuable review comments, we have added a new paragraph to the Introduction section. This paragraph provides a brief introduction to the work the researchers have done in constructing new chaos maps. Furthermore, we have added one chaotic map with modular operation (2D-MCS) to our comparative experiments. The relevant modifications are highlighted in yellow.
- The parameter range comparison for the LEs is a bit unfair. The parameter may range between 0 and 10, but in your map, the terms are exponential so basically we bifurcate between e^0=1 and e^10=22026.4657948, so understandably your map can obtain a much higher LE.
We would like to express our sincere thanks for the constructive comment on this point.
We completely agree with your comment, but please kindly allow us to explain something to you here. In fact, when the variable parameter is small, the LE values of our proposed 2D-ELMM are still relatively large. To highlight this, we have added two subfigures to Figure 2. That is, when the parameter range is [1,2], both LE values of 2D-ELMM are significantly larger than those of the other five maps. The relevant modifications are highlighted in yellow.
- In figure 4, I would also suggest including phase diagrams for the pairs (x(i-1), x(i)) so that we can see the connection between consecutive values of the states.
We would like to express our sincere thanks for your constructive comment and suggestion on this point.
According to your valuable review comments, we have added the corresponding phase diagrams to Figure 4. The relevant modifications are highlighted in yellow.
- Section 3.1, I do not understand, how we obtain the Q^0 series. this is clearly the values of the chaotic map, but how are the parameters chosen? Are they not plaintext dependent?
We would like to express our sincere thanks for your constructive comment and suggestion on this point.
Yes, we use the initial state values and control parameters of our proposed 2D-ELMM as the secret key. In our proposed CIES-DVEM, the secret key itself is plaintext-independent. At present, there are many encryption algorithms or schemes that use the hash value of the plaintext image as the secret key to generate the initial state values and control parameters of the chaotic system. Such a design is unreasonable and lacks practicability. First of all, when there are a large number of images to be encrypted, the encryption party and the decryption party need to constantly change the secret keys, which is not conducive to key management. Secondly, to encrypt each plaintext image, it is necessary to regenerate the chaotic sequence, which will obviously reduce the encryption efficiency.
Of course, in order to effectively resist various plaintext attacks, we introduce the hash value of the plaintext image in the encryption process. In other words, the equivalent keystreams of CIES-DVEM are plaintext-related.
- Also, in the same part, why is the range of the time series M+N+2*M*n? Since we have a 2d map, we can utilize each state, so the time series can be shorter. Please clarify.
We would like to express our sincere thanks for your constructive comment and suggestion on this point.
We are very sorry that we did not explain the relevant details. Actually, in CIES-DVEM, we do use each state value. Regarding this point, we have added the following explanation to Section 3.1:
Note that Q(0) is formed by interleaving the state values obtained by each iteration of 2D-ELMM, that is, Q(0)=(x1, y1, x2, y2, x3, y3, ...).
The relevant modifications are highlighted in yellow.
- Algorithm 2 and 3 are to me very unclear. Having the pseudocode is a good idea, but all I am seeing are some operations, many xors, and some sort of shuffling, which I am not sure what they do exactly in the big plan of the process. You need to clearly describe the operations in the text.
We would like to express our sincere thanks for your constructive comment and suggestion on this point.
According to your valuable suggestion, we have provided more detailed explanations of Algorithms 2 and 3. The relevant modifications are highlighted in yellow.
- I think this also needs some clarification, you specify the keys of the process, but from what I understand the hash of the image is also used. Is this hash also fed to the map parameters?
We would like to express our sincere thanks for your constructive comment and suggestion on this point.
According to your valuable suggestion, we have explained the use of the plaintext image hash value in more detail. The relevant modifications are highlighted in yellow.
- In figure 13, the 3d histogram in my opinion looks pointlessly complex. It would be much better if you simply included a 2d figure a histogram. What do the 3d axes even represent?
We would like to express our sincere thanks for your constructive comment and suggestion on this point.
In fact, more and more researchers now use this kind of diagram because it looks more intuitive and sophisticated. I distinctly recall encountering this kind of diagram for the first time in a paper authored by Hua Zhongyun. This paper is “Cross-plane colour image encryption using a two-dimensional logistic tent modular map.” In such a diagram, the x-axis and y-axis represent the coordinates of a pixel, and the z-axis represents the corresponding pixel value.
Therefore, we sincerely hope that you will allow us to use this kind of diagram.
- figure 15, try also cutting larger parts of the image, like 50% and 75%.
We would like to express our sincere thanks for your constructive comment and suggestion on this point. According to your valuable suggestion, we have increased the number of cut pixels to 25% and 50%, respectively. The relevant modifications are highlighted in yellow.
- "To encrypt an image of size M × N, CIES-DVEM only requires the employing of sequences with a total length of M + N + 2 × M × N, which is superior to many recent encryption schemes." I am not sure if this is really correct. Certainly, to perform xor, which every work does, we need M*N*8 bits to be computed, or as you say, M*N byte values. and then if we were to shuffle the rows and columns, we would again need M permutation values and N permutation values. So, M*N+M+N is a standardized kind of number. Saying that is it a short number is an exaggeration. Let's also not forget that you are hashing as well, this again requires some computation.
We would like to express our sincere thanks for your constructive comment and suggestion.
We completely agree with your comment, but it seems our focus is different. Please kindly allow us to explain something to you here. First of all, the length of the chaotic sequence we use is exactly M + N + 2 × M × N, and we can be 100% sure of this. Second, we only claim that the length of the sequence we use is relatively short. In fact, the chaotic sequences used in a large number of articles we have read are longer than this. Third, what we describe here is the length of the chaotic sequence used, which seems to have nothing to do with the hash value of the plaintext image.
- "Firstly, we optimized the method of utilizing the hash value, allowing for the chaotic sequences to be created beforehand and eliminating the need to constantly replace chaotic sequences" SO in your computational times reported, you do not include the time to generate the chaotic sequences? I believe this is unfair and wrong. The time for computing the chaotic sequence of length M*N+M+N must be included. And again i must ask, since the map is 2d, why not half of that length?
We would like to express our sincere thanks for your constructive comment and suggestion on this point.
Again, please kindly allow us to explain something to you. When we count the encryption time, we include the time required to generate the chaotic sequence for the first time. What we emphasize here is that when there are a large number of images, such as 100 images, that need to be encrypted, we only need to generate a chaotic sequence when encrypting the first image. Subsequent encryption does not need to regenerate the chaotic sequence, which is one of the advantages of not using the hash value as the secret key that we have been emphasizing.
- The rest of the simulation tests that are performed are all fine.
We would like to sincerely thank you for your efforts in reviewing our manuscript.
- Overall, I found some inaccuracies, and not a good enough description of the process. Unless these issues are correctly addressed, the work cannot proceed with publication. Please consider reproducibility of your results, and a very clear description of what is performed in each step. Should you feel it is required, even add a simple numerical example of a very small (10x10) matrix to make the process clear. For all these reasons, I would suggest major revisions for this work.
Once again, we would like to sincerely thank you for your efforts in reviewing our manuscript. Furthermore, for the problems and deficiencies you pointed out, we have also made serious revisions based on your valuable comments and suggestions. The relevant modifications are highlighted in yellow.
Finally, special thanks to your valuable comments and suggestions.
* * * * * * * * * * * * * * * * * * * * * * * * * * * * * * * * * * * * * * * * * * * *

Reviewer 2 Report
1. What are the limitations of proposed research work?
2. In what ways does image encryption support cryptography?
3. In Page.3, the authors claimed that "compared with 1D chaotic map, ....., can cause efficiency issues... ". Actually, these efficiency issue has been investigated in
"A parallel image encryption algorithm using intra bitplane scrambling. Mathematics and Computers in Simulation, 204, 71-88.", I suggest that the authors cite this paper to support their statement.
4. The author must demonstrate how image encryption is useful in real-life applications.
5. There is some inappropriate use, such as "4.2 Key Analysis->Key Space Analysis".
6. The authors should briefly introduce the method used for comparison in refs.[24, 33-36], so that the readers can know the difference between the proposed method and other methods.
Author Response
We would like to express our sincere thanks to the reviewers for their constructive comments and suggestions. Following the comments and suggestions, we have carefully revised our manuscript.
Below you will find our point-by-point responses to the reviewers' comments.
* * * * * * * * * * * * * * * * * * * * * * * * * * * * * * * * * * * * * * * * * * * * * *
Response to the comments of Reviewer 2
- What are the limitations of proposed research work?
We would like to express our sincere thanks for the constructive comment and suggestion on this point.
In the future, we will continue to further improve the efficiency and practicability of CIES-DVEM. First, we will try to introduce techniques such as compressed sensing, regions of interest, and neural networks. Secondly, we will further design a specific encryption step to ensure the plaintext correlation of the encryption process instead of relying on existing hash algorithms such as SHA-256. The relevant modifications are highlighted in yellow. Specifically, we have made the following modifications in the Conclusions section:
In the future, we will continue to enhance and optimize the proposed CIES-DVEM. For instance, a specific encryption step may be introduced to acquire plaintext features instead of relying on the SHA-256 hash function. Furthermore, our future research will try to introduce techniques such as compressed sensing, regions of interest, and neural networks.
- In what ways does image encryption support cryptography?
We would like to express our sincere thanks for the constructive comment and suggestion on this point.
At present, there are numerous studies on image encryption, and many researchers try to better protect image data by introducing new technologies and methods. In addition, lots of researchers have done a lot of cryptanalysis work on image encryption. We believe that these image encryption and cryptanalysis works are active explorations and expansions of traditional cryptography research.
- In Page.3, the authors claimed that "compared with 1D chaotic map, ....., can cause efficiency issues... ". Actually, these efficiency issue has been investigated in "A parallel image encryption algorithm using intra bitplane scrambling. Mathematics and Computers in Simulation, 204, 71-88.", I suggest that the authors cite this paper to support their statement.
We would like to express our sincere thanks for the constructive comment and suggestion on this point.
According to your valuable suggestion, we have cited the following three articles to support our argument:
- A parallel image encryption algorithm using intra bitplane scrambling.
- An image encryption scheme based on elementary and life-liked cellular automatons.
- A robustness-improved image encryption scheme utilizing Life-liked cellular automaton.
These three papers have deeply discussed the efficiency issues related to image encryption, which are of great reference value. The relevant modifications are highlighted in yellow.
- The author must demonstrate how image encryption is useful in real-life applications.
We would like to express our sincere thanks for the constructive comment and suggestion on this point.
According to your valuable suggestion, we have revised the first paragraph of the Introduction section in order to emphasize the important role of image encryption in real-life applications. The relevant modifications are highlighted in yellow.
- There is some inappropriate use, such as "4.2 Key Analysis->Key Space Analysis".
We would like to express our sincere thanks for the constructive comment and suggestion on this point.
According to your valuable suggestion, we have revised the title of Section 4.2 to Key Space Analysis. In addition, we have also carefully reviewed the full text in order to eliminate some other errors. The relevant modifications are highlighted in yellow.
- The authors should briefly introduce the method used for comparison in refs.[24, 33-36], so that the readers can know the difference between the proposed method and other methods.
We would like to express our sincere thanks for the constructive comment and suggestion on this point.
According to your valuable suggestion, we have provided a brief introduction to the methods employed in these articles in the second paragraph of the Introduction section. The relevant modifications are highlighted in yellow.
Finally, special thanks to your valuable comments and suggestions.

Round 2
Reviewer 1 Report
Dear editor and authors
The authors have properly addressed my questions.
As a final remark though, I would still argue that the 3d version of the histogram really is moot. Practically, this is like intead of using imshow(image) in matlab, you use surf(image) which plots it as a 3d surface, which you can then rotate in 3d and diplay it.
This really does not provide new information about pixel distribution, it is the same as visually inspecting the image with our eyes and saying 'this image has a lot of blue' etc. To me this is something that may be trending, but really not of much worth.
The histogram is on the other hand, a valuable statistical tool. It does represent how the pixels are distributed to the range of [0,255], and we can immediately observe if the distribution is uniform or not. So it is a valuable statistics tools based on theory. And it is a standard tool used for image analysis. I would suggest that you really consider going back to classic histogram diagrams in your future works.
Reviewer 2 Report
I have no further concerns.
Thank you for improving your work.